# Anthocyanins Modulation of Gut Microbiota to Reverse Obesity-Driven Inflammation and Insulin Resistance

**DOI:** 10.3390/nu17233727

**Published:** 2025-11-27

**Authors:** Caio Cesar Ruy, Tanila Wood dos Santos, Quélita Cristina Pereira, Marcelo Lima Ribeiro

**Affiliations:** Laboratory of Immunopharmacology and Molecular Biology, Sao Francisco University, Av. Sao Francisco de Assis, 218, Braganca Paulista 12916-900, SP, Brazil; ruyccaio@gmail.com (C.C.R.); tanilawood@gmail.com (T.W.d.S.); quelitapereirapa@gmail.com (Q.C.P.)

**Keywords:** obesity, anthocyanins, gut microbiota

## Abstract

Obesity has reached alarming proportions worldwide, becoming one of the most prevalent and critical public health challenges of the 21st century. Currently, there is great interest in studying the treatment of obesity with food-derived bioactive compounds, which have low toxicity and no serious adverse events compared to pharmacotherapeutic agents. Here, we review the benefits of anthocyanin-rich foods in preventing obesity, including antioxidant and anti-inflammatory effects, and in regulating the gut microbiota in preclinical models and human clinical trials. Evidence suggests that dietary anthocyanins may have anti-obesity effects and reduce the risk of chronic noncommunicable diseases by regulating gut health.

## 1. Introduction

Obesity has emerged as one of the most pressing public health challenges of the 21st century. Its prevalence has increased at an alarming pace over the past three decades, transforming from a regional concern to a global epidemic with profound clinical and socioeconomic consequences. According to the World Health Organization (WHO, 2024), by 2022 nearly one in eight individuals worldwide were living with obesity [1]. Since 1990, prevalence has more than doubled among adults and quadrupled among adolescents, reflecting complex interactions between urbanization, dietary transitions toward energy-dense ultra-processed foods, and reductions in physical activity. Global estimates indicate that approximately 2.5 billion adults were overweight in 2022, of whom 890 million were obese, corresponding to 43% of adults with overweight and 16% with obesity, figures that align with the Global Burden of Disease data [2,3,4,5,6,7,8,9].

Clinically, obesity is defined as excessive adiposity that increases health risk, often operationalized as a body mass index (BMI) ≥ 30 kg/m^2^. However, BMI alone fails to capture heterogeneity in fat distribution, metabolic profiles, and individual risk, highlighting the need for refined diagnostic tools based on body composition and metabolic markers [10]. Obesity is a systemic condition that affects virtually every organ and is strongly associated with type 2 diabetes (T2DM), hypertension, dyslipidemia, cardiovascular disease (CVD), metabolic dysfunction-associated steatotic liver disease (MASLD), and several cancers. Its pathophysiology extends beyond caloric imbalance, encompassing neuroendocrine regulation, chronic low-grade inflammation, alterations in gut barrier integrity and microbiota, and the interplay of genetic and epigenetic determinants [1,11,12,13,14,15,16,17,18,19,20].

The relentless rise in obesity prevalence underscores the inadequacy of current prevention and management strategies. Conventional approaches based solely on caloric restriction and exercise promotion often fail to ensure long-term weight maintenance, as they do not address underlying biological and environmental drivers. Consequently, there is a growing emphasis on integrative strategies that combine lifestyle modification, pharmacological therapy, and novel nutraceutical and microbiota-targeted interventions [21,22,23,24,25,26]. In particular, recent advances in understanding the gut–adipose–brain axis and the role of bioactive dietary compounds, such as anthocyanins, provide a promising framework for both mechanistic insights and translational therapeutic development determinants [12,18,27,28,29,30,31,32].

## 2. Energy Homeostasis and Neuroendocrine Regulation

Obesity results from a chronic positive energy balance governed by complex interactions between endocrine tissues and the central nervous system [33,34,35]. Energy homeostasis, the balance between energy intake and expenditure, is essential for body weight control. Resting metabolic rate, responsible for roughly 70% of total energy expenditure, is strongly associated with fat-free mass [35,36,37].

Food intake and energy expenditure are regulated by interlinked neural and hormonal pathways, with prominent communication between the central nervous system and peripheral tissues such as the gut and adipose tissue. Enteric sensory modalities detect nutrients and mechanical distension, triggering secretion of gut hormones (e.g., ghrelin, CCK, GLP-1, PYY) and autonomic afferent signaling that converge on brainstem nuclei and hypothalamic centers [38,39,40,41,42].

Adipose-derived signals, notably leptin and adiponectin, also modulate hypothalamic circuits (e.g., the arcuate nucleus) to influence feeding behavior and energy expenditure. In obesity, leptin signaling is commonly impaired (leptin resistance), while adiponectin levels are reduced [38,43,44,45,46]. Excess lipid deposition overwhelms lipid-handling pathways, causes endoplasmic reticulum and mitochondrial dysfunction, and increases reactive oxygen species generation; lipid intermediates such as ceramides activate pro-inflammatory kinases and foster an inflammatory adipokine profile that impairs insulin signaling [47,48,49,50,51,52,53,54].

Nutrient sensing by enteroendocrine cells stimulates secretion of hormones that regulate gastrointestinal motility and food intake. Ghrelin, an orexigenic peptide, increases appetite and can reduce physical activity, favoring weight gain. In contrast, GLP-1 is anorexigenic: it enhances insulin secretion, suppresses glucagon, slows gastric emptying, and acts centrally to reduce food intake [44,55,56,57,58,59,60,61,62,63].

These gut-derived signals interact with adipose-derived hormones and central networks to fine-tune feeding behavior and energy expenditure, and disruptions in these pathways contribute to the development and persistence of obesity.

## 3. Inflammation-Driven Mechanisms of Insulin Resistance in Obesity

There is growing evidence that obesity-associated inflammation originates at least in part from the gastrointestinal tract. High-fat diets impair distal intestinal epithelial barrier integrity, altering the expression of protective mucins and defensins [64,65,66,67,68,69,70,71,72].

Markers of inflammation with metabolic and endocrine functions include pro-inflammatory cytokines (IL-6, TNF-α, IL-1β, IL-18, IL-12), members of the IL-10 family, adipokines (e.g., adiponectin), chemokines (CCL2, CCL5, CXCL8, CXCL9, CXCL10), hepatocyte-derived acute-phase proteins (C-reactive protein—CRP, fibrinogen, serum amyloid A—SAA), downstream markers such as urinary microalbumin, and enzymes like COX-2 [43,73,74,75,76].

The inflammatory milieu in obesity, characterized by elevated cytokines and immune cell activation, promotes insulin resistance through multiple mechanisms. Inflammatory signals such as TNF alter kinase activities involved in insulin-mediated glucose uptake and impair insulin receptor signaling via aberrant phosphorylation patterns [77,78,79,80,81,82,83].

Elevated levels of free fatty acid intermediates activate NF-κB and protein kinase C (PKC), reinforcing inflammatory signaling and worsening insulin receptor dysfunction. Animal models show that suppression of pro-inflammatory kinase pathways can protect against insulin resistance [84,85,86,87].

Insulin signaling perturbations occur predominantly in liver and skeletal muscle, critically affecting glucose homeostasis. High hepatic concentrations of certain lipid intermediates, particularly diacylglycerol, correlate with hepatic insulin resistance, but tissue-specific interactions among nutrients, kinases and metabolic pathways remain incompletely understood. The regulation of energy metabolism involves complex interactions between peripheral hormones, such as ghrelin and leptin, and immune cells residing in metabolic tissues, particularly macrophages. Ghrelin, mainly produced in the stomach, acts through the GHSR receptor, promoting orexigenic effects and modulating intracellular pathways such as AMPK, in addition to exerting anti-inflammatory effects and favoring macrophage polarization toward the M2 phenotype, characterized by an anti-inflammatory profile and improved insulin sensitivity [88,89,90,91]. Conversely, leptin, secreted by adipocytes, acts on the ObR receptor via JAK2/STAT3 and PI3K/MAPK pathways, and under conditions of obesity and hyperleptinemia, induces a pro-inflammatory state that favors macrophage polarization toward the M1 phenotype, leading to the release of cytokines such as TNF-α, IL-6, and IL-1β, thereby contributing to the development of insulin resistance [92,93,94,95]. The polarization of macrophages into M1 and M2 represents a crucial point at the interface between immunity and metabolism: M1 macrophages rely predominantly on glycolytic metabolism, while M2 macrophages depend on fatty acid oxidation and AMPK activation, a kinase that acts as a cellular energy sensor regulating both inflammatory and metabolic pathways [94,96,97]. In obesity, there is increased infiltration of M1 macrophages in adipose tissue and a reduction in M2 cells, creating an inflammatory microenvironment that interferes with insulin signaling through inhibition of IRS-1 and reduced Akt phosphorylation [98]. AMPK activation has been associated with decreased M1 polarization, improved energy homeostasis, and reduced insulin resistance, highlighting its importance in modulating the NF-κB and JNK pathways and promoting the M2 phenotype [96,99]. Ghrelin activates AMPK in several tissues, including the pancreas and skeletal muscle, while leptin may either activate or inhibit this enzyme depending on the metabolic context, indicating that the balance between these hormonal pathways is essential for metabolic homeostasis. Furthermore, recent studies suggest that gap junctions, formed by connexins such as Cx43, play an important role in intercellular communication between macrophages and adipocytes, allowing the exchange of ions and metabolites that modulate the inflammatory response and insulin sensitivity [90]. Dysfunction of these junctions may amplify pro-inflammatory signaling and contribute to insulin resistance, whereas AMPK activation may preserve their function and limit the propagation of inflammation [100,101,102,103,104]. Thus, the combined dysregulation of ghrelin, leptin, and AMPK, together with impaired communication via gap junctions, promotes the shift in macrophages from the M2 to the M1 phenotype, thereby favoring insulin resistance. Therapeutic strategies aimed at restoring AMPK activation, balancing the effects of ghrelin and leptin, and preserving the integrity of gap junctions represent potential approaches for the treatment of metabolic and inflammatory syndromes associated with obesity [88,92,96,97,99].

## 4. Cardiometabolic and Hepatic Consequences of Insulin Resistance

Insulin resistance is a central metabolic disorder closely associated with numerous comorbidities, including type 2 diabetes, cardiovascular disease, hypertension and dyslipidemia. It is defined by the reduced capacity of tissues to respond appropriately to insulin, leading to hyperglycemia and compensatory hyperinsulinemia [105,106,107,108]. Approximately 80% of individuals with type 2 diabetes exhibit significant insulin resistance. Chronic hyperglycemia accelerates vascular damage and potentiates microvascular complications such as neuropathy and retinopathy [109,110,111,112,113,114].

Insulin resistance also predisposes to cardiovascular disease via endothelial dysfunction, adverse lipid profiles (elevated triglycerides, low HDL), and chronic low-grade inflammation that accelerate atherogenesis [115,116,117,118,119,120,121]. Visceral adiposity further increases cardiovascular risk through enhanced production of pro-inflammatory adipokines [122,123,124,125,126,127].

Insulin resistance is tightly linked to elevated blood pressure. Inflammatory pathway activation and endothelial dysfunction associated with insulin resistance contribute to abnormal blood pressure regulation. Studies indicate that insulin resistance can activate the sympathetic nervous system and the renin–angiotensin–aldosterone system, both crucial drivers of vasoconstriction and sodium retention in hypertension [116,128,129,130].

Insulin resistance is commonly accompanied by dyslipidemia: increased triglycerides, reduced HDL cholesterol, increased VLDL secretion, and decreased lipoprotein lipase activity, leading to lipid accumulation in liver and circulation [131,132,133,134,135,136,137,138]. Elevated uric acid associated with insulin resistance may also play a role in blood pressure elevation [139,140,141,142,143,144,145,146].

Lipid deposition in tissues, particularly the liver, can trigger MASLD and progress to cirrhosis. Hepatic steatosis arising from insulin resistance generates oxidative stress and pro-inflammatory cytokines, driving fibrosis and organ injury [147,148,149,150,151]. MASLD commonly coexists with metabolic syndrome, a cluster that includes abdominal obesity, hypertension, dyslipidemia and insulin resistance, with visceral adiposity being a key determinant of this syndrome [152,153,154,155,156,157,158].

## 5. Intestinal Barrier Integrity, Microbiota Shifts, and Their Role in Obesity-Related Inflammation

Recent evidence highlights the fundamental role of intestinal health and barrier integrity in metabolic regulation, inflammation, and the development of obesity-related conditions such as type 2 diabetes and cardiovascular disease. The intestinal barrier, formed by an epithelial cell layer connected through tight junctions, selectively allows nutrient absorption while blocking pathogens, toxins and other harmful substances [159,160,161,162,163,164,165,166,167].

In obesity, the intestinal barrier is often compromised, a condition commonly called ‘leaky gut,’ whereby increased permeability permits translocation of bacterial endotoxins such as lipopolysaccharide (LPS) into the circulation and triggers chronic innate immune activation [168,169,170,171,172,173,174,175]. This process contributes to the low-grade chronic inflammation observed in obesity and participates in insulin resistance and metabolic dysfunction [168,175,176,177,178,179,180,181].

Alterations in intestinal microbiota composition seen in obesity contribute to increased intestinal permeability and systemic inflammation. Specific microbial shifts, for example, changes in the abundance of Firmicutes relative to Bacteroidetes and increases in Proteobacteria, have been associated with higher endotoxin production and metabolic inflammation [182,183,184,185]. Microbiota-derived metabolites and structural molecules such as LPS engage host immune pathways, perpetuating a cycle of barrier dysfunction, inflammation and metabolic impairment [186,187,188,189,190,191,192,193,194,195].

LPS can activate the NF-κB immune pathway in the bloodstream. In combination with CD14, LPS acts as a ligand for Toll-like receptor (TLR) 4 [196]. This suggests that LPS translocation, caused primarily by high-fat diets, is related to obesity-induced low-grade systemic inflammation [177].

A chronic systemic accumulation of bacteria and LPS results in metabolic bacteremia and endotoxemia, respectively—pro-inflammatory processes characteristic of obesity and other metabolic syndrome phenotypes [153,197,198].

Microbial metabolites, such as short-chain fatty acids (SCFAs), secondary bile acids, and trimethylamine-*N*-oxide, may reflect gut microbiota activity and potentially predict obesity and insulin resistance risk. Therefore, increased SCFAs have been associated with decreased body weight, fat mass, waist circumference, fasting blood glucose, insulin resistance, and inflammation. Increased levels of secondary bile acids have been associated with decreased BMI, waist-to-hip ratio, fasting blood glucose, insulin resistance, and inflammation. Furthermore, elevated concentrations of trimethylamine-*N*-oxide were correlated with increased BMI, waist circumference, body fat percentage, fasting glucose, insulin resistance, blood pressure, inflammation, and oxidative stress [199,200,201].

## 6. Bioactive Phytochemicals and Microbiota-Targeted Interventions: Dual Strategies Against Obesity-Associated Inflammation

Interventions that modulate the gut microbiota, including dietary fiber enrichment, prebiotics, probiotics and symbiotics, have shown potential to restore barrier function, reduce systemic inflammation, and improve glucose homeostasis and cardiometabolic health [202,203,204,205,206,207,208]. Clinical and preclinical studies suggest that diets rich in fermentable fibers and low in saturated fats promote a healthier microbial community that produces SCFAs, enhances mucin production, and strengthens epithelial tight junctions [209,210,211,212,213,214,215].

Given the central role of inflammation in obesity-related disorders, exploration of anti-inflammatory strategies, including plant-derived natural compounds, has intensified. Flavonoids and other polyphenols present in fruits, vegetables and teas have shown capacity to modulate the microbiota, reinforce the epithelial barrier, and reduce chronic inflammation associated with obesity [216,217,218,219,220,221,222].

Natural compounds can upregulate tight junction proteins (occludin (Ocln), claudins (Cldn), Zonula occludens (ZO-1)) and inhibit pro-inflammatory pathways such as TLR4/NF-κB and NLRP3 inflammasome activation [223,224,225,226,227,228,229,230,231]. Examples include rhein, which restored occludin and ZO-1 distribution in LPS-stimulated IEC6 cells while reducing inflammatory cytokine secretion and inhibiting TLR4/NF-κB/NLRP3 activation [232], and glycyrol (a licorice derivative), which enhanced ZO-1 membrane localization through the GDNF/RET pathway [233]. A broad range of bioactive plant compounds, including curcumin, quercetin, resveratrol, naringenin, and ginsenosides, have demonstrated inhibition of central inflammatory pathways (e.g., TLR4/NF-κB), resulting in reduced expression of TNFα, IL-6 and IL-1β [227,234,235,236,237,238,239,240,241]. Flavonoids extracted from Amomum tsaoko attenuated inflammation in murine models of ulcerative colitis through inhibition of TLR4/NF-κB/NLRP3 signaling and simultaneous modulation of gut microbiota [242]. Ginger extracts have been reported to lower TNFα, reduce oxidative stress and restore tight junction proteins in experimental colitis models [243,244,245].

## 7. Anthocyanins: Chemical Characteristics, Bioavailability, Biological Effects, and Clinical Potential

Anthocyanins are water-soluble flavonoid pigments built on a C6–C3–C6 backbone and typically present in glycosylated or acylated forms (for example, aglycones such as cyanidin, delphinidin, pelargonidin, peonidin, petunidin and malvidin). Their chemical stability, encompassing color, pH dependence, light and temperature sensitivity, and sugar/acyl substituent patterns, significantly influences their presence in foods as well as their absorption and metabolism in the human body [246,247,248,249]. These compounds are abundant in red, blue or purple fruits and vegetables such as berries, grapes, red cabbage and eggplant, yet the actual dietary intake varies widely according to cultivar, harvest, processing and region. In terms of biological effects, growing meta-analyses of randomized controlled trials (RCTs) and cohort studies suggest promising benefits: one large meta-analysis of 44 RCTs and 15 cohort studies found that supplementation with purified anthocyanins significantly lowered LDL cholesterol and triglycerides, raised HDL, and habitual dietary consumption was associated with lower incidence of coronary heart disease and total cardiovascular disease [250,251,252,253]. A separate meta-analysis of 32 RCTs found that anthocyanins significantly improved fasting glucose, 2-h postprandial glucose, HbA1c, total cholesterol and LDL in cardiometabolic populations [254].

However, the bioavailability of anthocyanins remains a notable challenge: only a small fraction of ingested anthocyanins appear in the circulation in their intact form, as most are extensively metabolized in the gut and liver into glucuronide, sulfate and methylated derivatives [255,256,257]. Distribution to tissues in humans is poorly characterized, though animal studies indicate accumulation in liver, kidneys, brain and adipose tissue at low concentrations [258,259,260,261,262]. When considering clinical applicability, several factors must be addressed: the molecular form (glycosylated vs. aglycone), the food matrix (whole fruit versus extract vs. supplement), processing and digestion, interactions with other dietary components (fibers, fats, other polyphenols), inter-individual variability (microbiota composition, metabolism, transporters), and the need for standardized effective doses. For example, while evidence supports benefits, the heterogeneity in doses, formulations and populations limits translation to clinical practice [263,264,265,266].

## 8. Modulation of Gut Microbiota by Anthocyanins and Their Potential Synergistic Effects with Semaglutide and Tirzepatide

Anthocyanins have been shown to act as relevant modulators of gut microbiota, promoting beneficial bacteria such as Bifidobacterium spp. and Bacteroides, increasing short-chain fatty acid (SCFA) production (acetate, propionate, butyrate), reducing intestinal pH and permeability, and improving tight junction protein expression and villi morphology in animal models [267,268,269,270]. Concurrently, GLP-1 receptor agonists such as Semaglutide and dual GIP/GLP-1 receptor agonists like Tirzepatide have been demonstrated to modulate the gut microbiota in animal models, increasing the abundance of Akkermansia muciniphila, Faecalibaculum, Allobaculum, and Muribaculaceae, while improving intestinal barrier integrity and reducing inflammation [271,272,273,274,275]. Given that both anthocyanins and these pharmacological agents converge on the gut microbiota pathway, it is plausible that anthocyanin intake may potentiate or modify their effects by promoting beneficial bacteria that are also favored by the drugs, increasing SCFA production which activates GPR43/GPR41 receptors, and enhancing intestinal integrity, thereby reducing metabolic endotoxemia. While direct clinical evidence of this synergy is lacking, mechanistic data suggest potential complementary effects, particularly in the context of obesity or type 2 diabetes. Most current evidence derives from animal or in vitro studies; human data remain limited, and factors such as anthocyanin dose, formulation, duration, and individual microbiota profiles may influence outcomes [276,277,278,279]. Well-designed clinical trials are therefore urgently needed to evaluate anthocyanin supplementation in patients treated with Semaglutide or Tirzepatide, monitoring microbiota composition, SCFA levels, intestinal barrier function, and therapeutic response, to explore their translational and synergistic potential in a safe and standardized manner.

## 9. Anthocyanins: In Vitro, In Vivo and Clinical Evidence

Anthocyanins, water-soluble flavonoid pigments abundant in berries and colored fruits, have shown antioxidant, hypoglycemic and hypolipidemic effects in vitro and in animal models (Table 1) [280,281,282]. For example, anthocyanins isolated from blueberry (*Vaccinium virgatum*, ‘Garden blue’) increased glucose uptake and reduced lipid accumulation in HepG2 cells exposed to high glucose and oleic acid, suggesting therapeutic potential for obesity and diabetes, though animal validation is still needed [283].

A randomized clinical trial evaluated supplementation with blueberry anthocyanins combined with prebiotics (rice bran fiber and Jerusalem artichoke) in patients with type 2 diabetes. After 60 days, participants showed reductions in fasting glucose, HbA1c and LDL cholesterol, and improvement in estimated glomerular filtration rate; however, markers of oxidative stress, inflammation and cardiorespiratory fitness were unchanged, indicating that mechanistic pathways require further exploration [284].

Other studies have reported cardiometabolic benefits of anthocyanin-rich extracts (e.g., *Aronia melanocarpa*, pomegranate, mulberry) in animal models and small human trials, including reductions in LDL cholesterol, triglycerides, blood pressure and postprandial endotoxemia, though results are heterogeneous across studies [254,285,286,287,288,289,290,291].

Mechanistically, anthocyanins and related polyphenols interact with multiple molecular targets: they activate AMPK to inhibit lipogenesis and enhance fatty acid oxidation, stimulate Nrf2-driven antioxidant defenses, attenuate NF-κB-mediated inflammation, and modulate transcription factors controlling lipid and cholesterol homeostasis (LXR, PPARα, PPARγ, C/EBPα, SREBP-1c) [292,293,294,295,296,297,298,299,300,301,302,303,304,305,306].

In recent years, growing evidence has shown that anthocyanidins, particularly cyanidin-3-O-glucoside (C3G), exert beneficial immunometabolic effects through the modulation of insulin signaling, leptin activity, and macrophage polarization. For example, C3G has been reported to activate AMPK, inhibit the NF-κB pathway, and reduce pro-inflammatory cytokine production, thereby promoting the shift in macrophages from the M1 to the M2 phenotype, consistent with anti-inflammatory effects and enhanced insulin sensitivity [307,308,309]. Recent reviews also indicate that anthocyanins regulate leptin signaling by increasing ObR receptor expression and activating the JAK2/STAT3 and PI3K/Akt pathways, which improves leptin sensitivity and reduces insulin resistance [218]. Network-based analyses of bioactive compounds further highlight anthocyanins as critical nodes capable of shifting an obesogenic, pro-inflammatory state toward a healthier phenotype by promoting the recruitment of M2 macrophages in adipose tissue [309,310,311]. Although direct studies on ghrelin remain limited, the energy-rebalancing effects mediated by AMPK activation may indirectly influence ghrelin signaling, as ghrelin interacts with cellular energy pathways that shape macrophage polarization [96,312,313]. This combination of effects, enhanced insulin signaling, normalization of leptin activity, and a macrophage shift toward the M2 phenotype, suggests that anthocyanidins hold considerable therapeutic potential for mitigating chronic inflammation associated with obesity and insulin resistance (Figure 1).

Specific compounds such as cyanidin-3-O-glucoside (C3G) can stimulate GLP-1 secretion from intestinal L cells via PPARβ/δ–β-catenin–TCF-4 signaling, thereby enhancing insulin secretion and improving glycemic control [314,315,316]. C3G has also demonstrated protective effects on pancreatic β-cells against lipotoxic stress by attenuating ER stress and CHOP-mediated apoptosis [317]. Polyphenols act on integrated metabolic and signaling pathways, supporting their candidacy as nutraceutical modulators of glucose and lipid metabolism.

**Table 1 nutrients-17-03727-t001:** Therapeutic effects of anthocyanins in experimental and clinical models.

Anthocyanins	Food Source	Main Metabolic Effects	Molecular Mechanisms	Evidence (In vitro/In vivo/Human)
Cyanidin-3-O-glucoside (C3G)	Blueberry, Blackcurrant	↑ GLP-1, ↑ Insulin, ↓ Lipotoxicity	PPARβ/δ–β-catenin–TCF-4, ER stress attenuation	In vitro [280,314,317] In vivo [282,289] Human [281,284]
Delphinidin	Blackcurrant, Elderberry	↓ Inflammation, ↑ Antioxidant status	NF-κB inhibition, Nrf2 activation	In vitro [283,303,304,305] In vivo [283,288,302] Human [306]
Malvidin	Blueberry, Grape	↓ Lipogenesis, ↓ Cholesterol	AMPK activation, PPARα/γ modulation	In vitro [301] In vivo [293,294,301] Human [292]
Pelargonidin	Strawberry, Raspberry	↓ Postprandial glycaemia, ↓ LDL	NF-κB, LXR modulation	In vitro [296,297] In vivo [290,291,296,297,298] Human [254]
Petunidin	Blueberry, Purple star apple	↑ Fat acid oxidation, ↓ TG	AMPK/Nrf2, SREBP-1c inhibition	In vitro [299,301] In vivo [300,301] Human [285,286]

PPARβ/δ—Peroxisome Proliferator-Activated Receptor Beta/Delta; TCF—T-Cell Factor; ER—Estrogen Receptor; NF-κB—Nuclear Factor kappa-Light-Chain-Enhancer of Activated B Cells AMPK—AMP-Activated Protein Kinase; PPARα/γ—Peroxisome Proliferator-Activated Receptor Alpha/Gamma; LXR—Liver X Receptor; LDL—Low-Density Lipoprotein; TG—Triglycerides; AMPK/Nrf2—AMP-Activated Protein Kinase/Nuclear Factor Erythroid 2–Related Factor 2; SREBP-1c—Sterol Regulatory Element-Binding Protein 1c. ↑—Induction; ↓—Repression.

## 10. Anthocyanins, Microbiota and Gut–Liver Interactions

Anthocyanins influence gut microbial composition and function, promoting SCFA production and favoring genera such as *Akkermansia* and *Parabacteroides* while reducing the *Firmicutes/Bacteroidetes* ratio in some models [276,318,319,320,321,322]. These changes can improve intestinal barrier function, reduce endotoxemia and modulate hepatic metabolism via activation of the Nrf2/Keap1 pathway and SCFA-mediated signaling [323,324].

Microbial biotransformation of flavonoids generates metabolites (e.g., cyanidin derivatives, γ-valerolactones) that contribute to antioxidant and metabolic effects observed in vitro and in vivo [325,326,327,328,329,330]. The bidirectional interaction between flavonoids and microbiota, where polyphenols modulate microbiota composition and microbiota metabolizes polyphenols into bioactive compounds, sustains a beneficial cycle for barrier integrity and host metabolism [331,332,333,334,335,336,337]. Microbiota-mediated metabolism of anthocyanins amplifies their systemic effects and constitutes an important mechanism linking diet to metabolic health.

Although a substantial body of preclinical research supports metabolic benefits of anthocyanins and other phytochemicals, most of the evidence arises from in vitro or animal models. Well-designed human trials are required to determine optimal doses, formulations, safety profiles and bioavailability in clinically relevant populations.

Heterogeneity in food matrices, processing, individual metabolic phenotypes and microbiota composition influences polyphenol bioavailability and response to supplementation. This variability necessitates personalized and integrative approaches when recommending these compounds in clinical practice (Figure 2).

## 11. Concluding Remarks

Recognized for their anti-inflammatory, antioxidant, and immunomodulatory properties, polyphenols are increasingly recognized for their potential interconnection with the complex dynamics of the gut microbiota in the pathogenesis of various diseases. Furthermore, there is a growing focus on how microbiota can initiate epigenetic adjustments, with polyphenolic compounds potentially holding the key to counteracting these obesity-related epigenetic alterations.

The evidence reviewed here supports a model in which obesity-driven dysregulation of energy balance, gut barrier dysfunction, and chronic low-grade inflammation converge to produce insulin resistance and cardiometabolic disease. Natural compounds, particularly anthocyanins and other polyphenols, exert pleiotropic actions that target inflammation, barrier integrity, microbiota composition, and metabolic signaling pathways. These intricate interactions represent a promising advance in addressing dysbiosis and its associated epigenetic nuances. This area offers potential for innovative nutritional interventions aimed at combating the complex biological effects of obesity.

## Figures and Tables

**Figure 1 nutrients-17-03727-f001:**
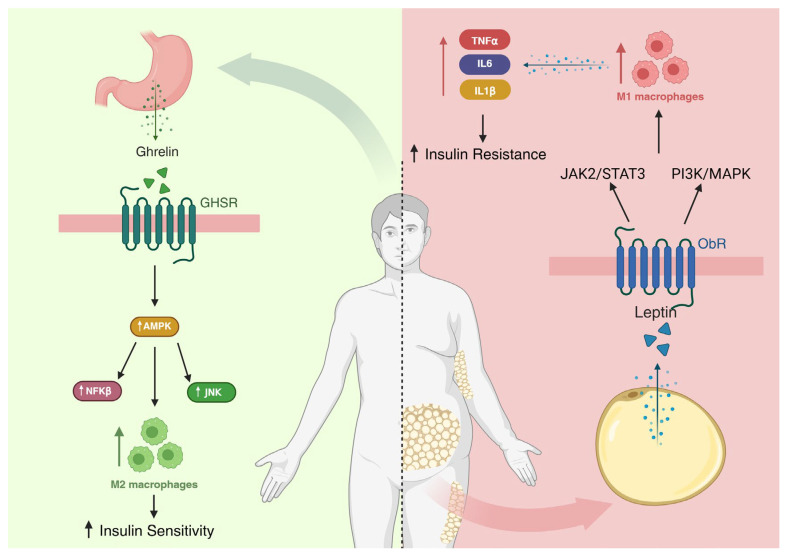
Schematic representation of the interplay between ghrelin and leptin in the immunometabolic regulation of insulin sensitivity. Ghrelin, mainly secreted by the stomach, acts through the GHSR receptor to activate AMPK signaling, promoting anti-inflammatory responses and macrophage polarization toward the M2 phenotype, which enhances insulin sensitivity. In contrast, leptin, secreted by adipocytes, signals via the ObR receptor through the JAK2/STAT3 and PI3K/MAPK pathways, leading to M1 macrophage polarization and the release of pro-inflammatory cytokines (TNF-α, IL-6, and IL-1β), contributing to insulin resistance. The balance between ghrelin- and leptin-mediated pathways, together with proper AMPK activation, is essential for maintaining metabolic homeostasis and preventing chronic inflammation in metabolic tissues. GHSR—Growth Hormone Secretagogue Receptor; AMPK—AMP-Activated Protein Kinase; NF-κB—Nuclear Factor kappa-Light-Chain-Enhancer of Activated B Cells; JNK—c-Jun N-Terminal Kinase; M2—Alternatively Activated (Anti-inflammatory) Macrophage; M1—Classically Activated (Pro-inflammatory) Macrophage; TNFα—Tumor Necrosis Factor Alpha; IL-6—Interleukin 6; IL-1β—Interleukin 1 Beta; JAK2/STAT3—Janus Kinase 2/Signal Transducer and Activator of Transcription 3; PI3K/MAPK—Phosphoinositide 3-Kinase/Mitogen-Activated Protein Kinase; ObR—Leptin Receptor. ↑—Induction; ↓—Repression.

**Figure 2 nutrients-17-03727-f002:**
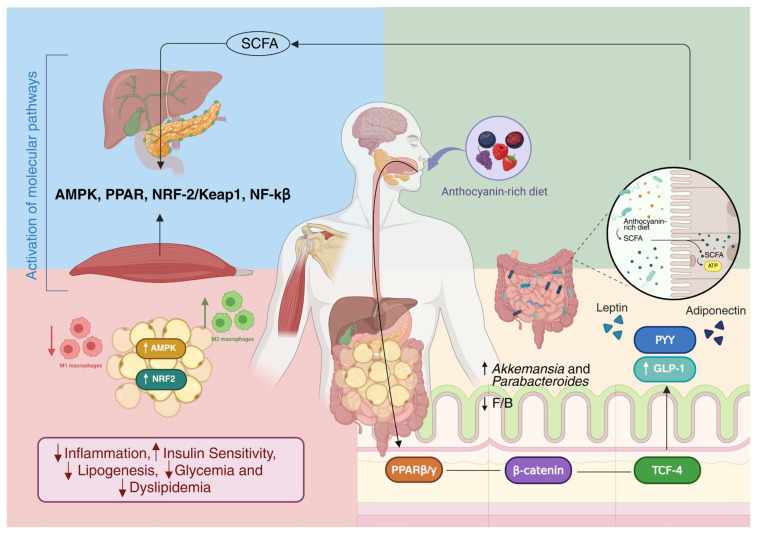
Integrated mechanisms of action of anthocyanins in obesity. An anthocyanin-rich diet modulates the gut microbiota, promoting the growth of beneficial bacterial genera such as Akkermansia and Parabacteroides and enhancing the production of short-chain fatty acids (SCFAs), which strengthen the intestinal barrier. SCFAs and anthocyanins stimulate the secretion of gut hormones (GLP-1 and PYY) and adipokines (leptin and adiponectin), contributing to improved metabolic signaling. In peripheral tissues such as the liver, muscle, and pancreas, anthocyanins activate key molecular pathways, including AMPK, PPARγ, NRF2/Keap1, and NF-κB, leading to reduced inflammation, increased insulin sensitivity, and decreased lipogenesis, glycemia, and dyslipidemia. [276,292,293,294,295,296,297,298,299,300,301,302,303,304,305,306,314,318,319,320,321,322,323,324]. PPAR—Peroxisome Proliferator-Activated Receptor; NF-κB—Nuclear Factor kappa-light-chain-enhancer of Activated B Cells; AMPK—AMP-Activated Protein Kinase; NRF-2/Keap1—Nuclear Factor Erythroid 2–Related Factor 2/Kelch-like ECH-Associated Protein 1; F/B—Firmicutes/Bacteroidetes; PYY—Peptide YY; GLP-1—Glucagon-Like Peptide-1; TCF-4—Transcription Factor 4; ATP—Adenosine Triphosphate; SCFA—Short-Chain Fatty Acids; M1—Classically Activated (Pro-inflammatory) Macrophage; M2—Alternatively Activated (Anti-inflammatory) Macrophage. ↑—Induction; ↓—Repression.

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
