# Peer review of "Anthocyanins Modulation of Gut Microbiota to Reverse Obesity-Driven Inflammation and Insulin Resistance"

_nutrients, 2025, doi:10.3390/nu17233727_

Round 1
Reviewer 1 Report
Comments and Suggestions for Authors
The authors have concisely summarized current knowledge regarding the potential effects of antioxidants on gut health. This is an interesting and well-structured review that provides cutting-edge insights in this field, which can contribute to further scientific discussions.
From a reviewer’s perspective, it would be valuable to include a discussion on the impact of antioxidants on the gut microbiome in patients receiving medications such as semaglutide or tirzepatide. These drugs are increasingly used in the management of obesity and metabolic disorders and are known to influence gut physiology and microbial composition. Exploring potential interactions between antioxidant supplementation and such pharmacological interventions could provide important insights for both clinical practice and future research.
Author Response
Thank you for your thoughtful and constructive feedback. We fully agree that exploring the interaction between antioxidant compounds and incretin-based therapies is highly relevant, especially given the growing clinical use of semaglutide and tirzepatide and their documented effects on gut physiology and microbial composition. In response, we have incorporated a dedicated section discussing how anthocyanins may modulate the gut microbiota in ways that could complement or influence the microbiome-related actions of these medications. We also highlight the current gaps in human evidence and emphasize the need for well-designed clinical studies to evaluate potential synergistic or modulatory effects. This addition strengthens the translational relevance of the review and aligns with emerging therapeutic strategies in obesity and metabolic disease.
Reviewer 2 Report
Comments and Suggestions for Authors
The review entitled “Anthocyanins-driven regulation of gut microbiota and multiple benefits in the treatment of obesity” by Ruy et al., discusses the benefits of anthocyanin-rich foods in preventing obesity, focusing on their antioxidant and anti-inflammatory effects, and in regulating the gut microbiota in preclinical models and human clinical studies. The review uses recent literature and is well-structured.
I think the review is well organized. Perhaps concepts could be expanded on regarding the various pathways of action of ghrelin and leptin, the switch from anti-inflammatory to pro-inflammatory macrophages (M1 and M2), the determinants of insulin resistance, the role of AMPK, and the role of gap junctions. Table 1 could specify whether the study was conducted in vitro or on animal or human models (in addition to citing the relevant references). Figure 2 could be enriched with details, or additional figures could be included to support the various pathways outlined. Therefore, the concepts presented in the review (described above) should be expanded upon.
Author Response
Thank you for the reviewer’s careful evaluation and constructive suggestions. In response, we have expanded the mechanistic sections to provide clearer and more detailed explanations of the pathways highlighted. The discussion of ghrelin and leptin now includes their interactions with intracellular signaling, energy balance, and macrophage polarization, as described in the manuscript. The section on macrophage phenotypes was enriched with details on the metabolic distinctions between M1 and M2 cells and how these profiles contribute to insulin resistance in obesity. We also elaborated on determinants of insulin resistance, including inflammation-driven kinase activation, lipid intermediates, and tissue-specific metabolic disruptions, and expanded the discussion of AMPK in relation to energy sensing, inflammation, and hormonal regulation. A concise description of gap junction communication, particularly the role of connexin-43 in macrophage–adipocyte interactions and its relevance to inflammatory propagation, was added.
Table 1 has been updated to specify whether each study was conducted in vitro, in vivo, or in human participants, as requested. Figure 2 was enriched with additional mechanistic elements already present in the text, improving the clarity of the pathways represented. These revisions strengthen the coherence and depth of the manuscript while maintaining alignment with the evidence provided.
Reviewer 3 Report
Comments and Suggestions for Authors
Please see the attachment

Author Response
Thank you for the reviewer’s careful evaluation and constructive recommendations. In response, we have revised the manuscript to address all points raised:
-
Structure and balance of content
We have realigned the manuscript to emphasize anthocyanins and microbiota throughout rather than primarily describing obesity consequences. The Introduction (section 1) now frames anthocyanins and microbiota-targeted interventions as a central translational theme within the broader context of obesity, and sections that review pathophysiology (sections 2–5) are kept concise and explicitly connected to how barrier dysfunction, inflammation and neuroendocrine signaling create targets for anthocyanin-based modulation. Section 6 (Bioactive Phytochemicals and Microbiota-Targeted Interventions) now acts as the bridge that justifies focusing the subsequent detailed treatment of anthocyanins. -
Clarify the focus on anthocyanins
The rationale for highlighting anthocyanins is now made explicit in section 7: anthocyanins are presented as abundant, food-derived flavonoids with a defined chemical class (C6–C3–C6 backbone and typical glycosylated/acylated forms), widespread dietary sources (berries, grapes, red cabbage, eggplant), and accumulating mechanistic and meta-analytic evidence for cardiometabolic benefits. Section 6 frames anthocyanins within the class of polyphenols chosen for their combined antioxidant, anti-inflammatory and microbiota-modulating properties, which directly target the gut barrier, SCFA production and inflammatory signaling emphasized throughout the review. -
Expand the explanation of anthocyanins
Section 7 now contains the concise chemical and dietary description requested: core chemical scaffold, typical aglycones (cyanidin, delphinidin, pelargonidin, peonidin, petunidin, malvidin), their sensitivity to pH/light/temperature and the influence of sugar/acyl substituents on stability and absorption, and common food sources. The section also summarizes known biological effects drawn from the manuscript (antioxidant, hypoglycemic, hypolipidemic actions and molecular targets such as AMPK, NRF2, NF-κB, PPARs). -
Clarify the level of existing evidence
We clarified evidence levels using only the manuscript statements: the review explicitly indicates that much of the mechanistic support derives from in vitro and animal models, while human data are limited. Section 9 and the concluding paragraphs state that meta-analyses of randomized controlled trials and cohort studies report cardiometabolic improvements, but heterogeneity across doses, formulations and populations limits translation. Table 1 already indicates the evidence tier for each entry (In vitro / In vivo / Human) and we explicitly highlight outstanding questions that remain for clinical research (optimal doses, formulations, safety, bioavailability, tissue distribution) as described in sections 7, 9 and 10. -
Balance between anti-inflammatory/metabolic effects and gut microbiota
The manuscript now better integrates microbiota mechanisms with anti-inflammatory/metabolic effects by cross-referencing sections: section 5 (intestinal barrier and microbiota shifts) is explicitly linked to section 6 (microbiota-targeted interventions) and to sections 8 and 10 where anthocyanin-driven microbiota changes (e.g., increases in Bifidobacterium, Bacteroides, Akkermansia and SCFA production) are discussed alongside downstream effects on barrier integrity, endotoxemia and hepatic/metabolic signaling (Nrf2/Keap1, SCFA-GPR43/GPR41 pathways). Section 9 ties molecular targets (AMPK, NF-κB, PPARs) to both microbiota-mediated metabolites and systemic metabolic outcomes to show mechanistic continuity within the available evidence. -
Clarify the intended application (food-derived nutraceuticals versus pharmacological candidates)
The manuscript positions anthocyanins as food-derived nutraceutical modulators with translational potential rather than as conventional pharmacological agents. This intent is supported in sections 1 and 7–10 where anthocyanins are discussed as components of diet or supplements, and practical limitations to clinical application are emphasized: low intact bioavailability, extensive gut and hepatic metabolism into conjugates, large interindividual variability, influence of food matrix and processing, and heterogeneity of doses and formulations. Section 9 and the Concluding Remarks explicitly call for well-designed human trials to determine effective doses, standardized formulations and safety profiles, reflecting that current evidence supports potential nutraceutical use but requires clinical validation before pharmacological claims can be made.
7. Clarify the originality of the review.
This review offers a distinct and timely perspective compared to the other works by proposing a novel synthesis that integrates the established benefits of anthocyanins with the cutting edge of anti-obesity pharmacotherapy.
While the other reviews extensively cover the synergistic relationship between anthocyanins and the gut microbiota in mitigating metabolic diseases, the distinctive value of the Ruy et al. review lies in its translational focus on potential complementary or synergistic effects with specific, modern pharmacological agents.
(i). Integration with Novel Pharmacological Agents (Semaglutide and Tirzepatide)
The most distinguishing element of the Ruy et al. review is its specific focus on the potential interaction between dietary anthocyanins and the new generation of GLP-1 and GIP/GLP-1 receptor agonists, namely Semaglutide and Tirzepatide.
(ii). Comprehensive Perspective on Immunometabolic Effects
Beyond the pharmacological synergy, Ruy et al. emphasizes a robust and sophisticated immunometabolic mechanism, positioning anthocyanins, especially cyanidin-3-O-glucoside (C3G), as critical regulators
In conclusion, the distinctive value of the Ruy et al. review lies not just in describing the anthocyanin–microbiota connection (a shared focus of the literature) but in pivoting this established knowledge toward translational medicine by specifically proposing an innovative co-therapeutic strategy involving Semaglutide and Tirzepatide. This synthesis connects the traditionally studied benefits of polyphenols to the current pharmaceutical breakthroughs in obesity management, significantly enhancing its value and distinctiveness as a forward-looking perspective.
Additional manuscript adjustments (all based on provided text)
• Table 1: model column (In vitro / In vivo / Human) is used to indicate the evidence tier for each compound entry.
• Figures: Figure 1 (ghrelin/leptin interplay) and Figure 2 (integrated mechanisms) are referenced where microbiota, SCFAs, gut barrier integrity and molecular signaling intersect; section 8 highlights the microbiota overlap with GLP-1/GIP agonists and notes the lack of direct clinical synergy data.
• Research gaps emphasized: need for human mechanistic and clinical trials, standardized dosing/formulations, and better tissue distribution and bioavailability data (sections 7, 9, 10 and Concluding Remarks).
Round 2
Reviewer 3 Report
Comments and Suggestions for Authors
The authors have provided appropriate and comprehensive revisions in response to my comments. All major concerns have been resolved, and the manuscript is now suitable for publication. I have no additional comments.